# From Generalist to Specialist: Incorporating Domain-Knowledge into Flamingo for Chest X-Ray Report Generation

**Raphael Stock**[1]                                              RAPHAEL.STOCK@DKFZ-HEIDELBERG.DE
**Stefan Denner**[1]                                              STEFAN.DENNER@DKFZ-HEIDELBERG.DE
**Yannick Kirchhoff**[1]                                       YANNICK.KIRCHHOFF@DKFZ-HEIDELBERG.DE
**Constantin Ulrich**[1]                                       CONSTANTIN.ULRICH@DKFZ-HEIDELBERG.DE
**Maximilian R. Rokuss**[1]                               MAXIMILIAN.ROKUSS@DKFZ-HEIDELBERG.DE
**Saikat Roy**[1]                                                  SAIKAT.ROY@DKFZ-HEIDELBERG.DE
**Nico Disch**[1]                                                  NICO.DISCH@DKFZ-HEIDELBERG.DE
**Klaus Maier-Hein**[1,2]                                     K.MAIER-HEIN@DKFZ-HEIDELBERG.DE

[1] *Division of Medical Image Computing, German Cancer Research Center, Heidelberg, Germany*

[2] *Pattern Analysis and Learning Group, Heidelberg University Hospital, Germany*

## Abstract

Automating the generation of accurate and reliable radiological reports from chest X-ray images represents a significant challenge in medical image computing. In this context, Vision-Language Models (VLMs), particularly the Flamingo architecture which achieves state-of-the-art performance across various vision-language tasks, offers promising solutions. This study evaluates the effectiveness of OpenFlamingo and its medical adaptation MedFlamingo, a version further pre-trained on medical data, in generating radiological reports. Our evaluation compares the zero-shot capabilities of OpenFlamingo and MedFlamingo against fine-tuning and training from scratch. Our results demonstrate that fine-tuning consistently boosts model performance, with fine-tuned MedFlamingo outperforming its OpenFlamingo counterpart. Moreover, while training Flamingo from scratch does not match the efficacy of fine-tuning, it nevertheless surpasses zero-shot performance. This study underscores the potential of domain-specific fine-tuning in enhancing automated radiological report generation, paving the way for more accurate and efficient diagnostic workflows.

**Keywords:** Vision-language model, chest X-ray, report generation

## 1. Introduction

In recent years, Vision-Language Models (VLMs) have emerged as powerful tools with promising applications across various domains (Liu et al., 2024; Achiam et al., 2023), including medicine (Chen et al., 2024; Wu et al., 2023; Hyland et al., 2023). Notably, the interpretation of chest X-ray images, the most commonly performed diagnostic examination in the USA (Iyeke et al., 2022), presents a significant area for the application of these technologies. This study evaluates the report generation capabilities of different versions of a state-of-the-art VLM architecture called Flamingo (Alayrac et al., 2022), focusing on their ability to produce findings from chest X-ray images. In particular, we compare the zero-shot capabilities of two open-source Flamingo models from the natural and medical domain and fine-tune them on the MIMIC-CXR dataset (Johnson et al., 2019). We highlight the benefits of different pre-training pathways for specialized medical tasks.

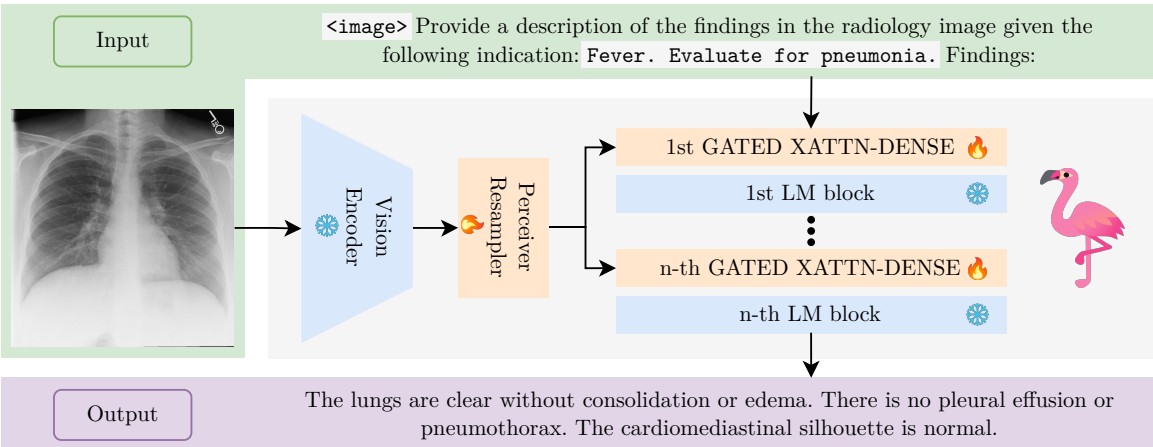

Figure 1: Flamingo architecture and report finding generation task. In this paper, we only train the Flamingo-specific modules which are indicated with a 🔥.

## 2. Methods

**Data**  We utilize the MIMIC-CXR dataset (Johnson et al., 2019) for our experiments. It consists of 227k studies for over 65k patients. Using the official MIMIC-CXR codebase[1], we extract the indication and findings sections of the written reports. We use the provided MIMIC-CXR dataset splits and filter for studies with frontal view. Additionally, we only utilize the first study of a patient to prevent hallucinations (Hyland et al., 2023) originating from longitudinal data. We provide the resulting splits file in the linked repository[2].

**Models & training**  OpenFlamingo (Awadalla et al., 2023) is used as the starting point for the evaluation. It uses CLIP's ViT (Radford et al., 2021) as vision encoder and LLaMA (Touvron et al., 2023) as language model. The embeddings from the vision transformer are first passed through a Perceiver Resampler and then fed into the language model via Gated Cross-Attention layers (cf. Figure 1). Originally, OpenFlamingo was trained on the two open-source web-scraped image-text datasets LAION-2B (Schuhmann et al., 2022) and MMC4 (Zhu et al., 2024). The domain-specific medical model, MedFlamingo (Moor et al., 2023), was obtained by further fine-tuning OpenFlamingo on data originating from over 4500 medical text books.

In this work, we further fine-tune both OpenFlamingo and MedFlamingo checkpoints on the previously described subset of the MIMIC-CXR dataset. Following the training scheme of Flamingo, we only train the weights of the Perceiver Resampler and the Gated Cross-Attention layers.  As in Hyland et al. (2023) our input prompt is "`<image>` `Provide a description of the findings in the radiology image given the following indication:` `<INDICATION> Findings:` " if an indication exists, otherwise if the indication is not provided the input prompt is "`<image> Provide a description of the findings in the radiology image. Findings:` ".  Here, `<image>` is the image token and `<INDICATION>`

---

1. https://github.com/MIT-LCP/mimic-cxr
2. https://doi.org/10.5281/zenodo.10953053

Table 1: Zero-shot ( 0 ) and fine-tuned ( 🔥 ) performance of different Flamingo ( 🦩 ) models on the report generation task on our MIMIC-CXR test split. For all metrics a higher score is better except for the RadCliQ metric, which is indicated by an ↓.

| | Metric / Model | Lexical | | | | | Semantic | Clinical | | |
|---|---|---|---|---|---|---|---|---|---|---|
| | | BLEU-1 | BLEU-2 | BLEU-4 | METEOR | ROUGE-L | BERTScore | RadGraphF1 | CheXbert | RadCliQ (↓) |
| 0 | Open 🦩 | 0.106 | 0.032 | 0.000 | 0.072 | 0.073 | -0.176 | 0.041 | 0.186 | 5.677 |
| | Med 🦩 | 0.148 | 0.071 | 0.012 | 0.123 | 0.122 | 0.122 | 0.097 | 0.217 | 4.685 |
| | 🦩 from scratch | 0.191 | 0.140 | 0.044 | 0.210 | 0.222 | 0.347 | **0.218** | 0.291 | 3.773 |
| 🔥 | Open 🦩 | 0.204 | 0.146 | 0.047 | 0.218 | **0.228** | **0.360** | 0.216 | 0.304 | 3.714 |
| | Med 🦩 | **0.216** | **0.153** | **0.048** | **0.220** | **0.228** | 0.355 | **0.218** | **0.327** | **3.683** |

will be replaced by the report's indication. We train the model on the task of next token prediction. Additionally, we train from scratch using the OpenFlamingo architecture and randomly initializing the Flamingo-specific modules. We select the checkpoint with minimal validation loss for all trained models, which is reached after three epochs for OpenFlamingo and MedFlamingo and after six epochs for the model trained from scratch.

**Evaluation** For evaluation, we consider lexical and radiology-specific metrics, as well as the BERTScore (Zhang et al., 2019). For lexical metrics we compare BLEU-{1,2,4} (Papineni et al., 2002), METEOR (Banerjee and Lavie, 2005) and ROUGE-L (Lin, 2004) using huggingface's evaluate library[3]. The radiology-specific metrics include RadGraphF1 (Delbrouck et al., 2022), CheXbert (Smit et al., 2020) and RadCliQ (Yu et al., 2023)[4].

## 3. Results and Discussion

The results are summarized in Table 1. Both fine-tuned models outperform their zero-shot counter parts, indicating that generalist models benefit from fine-tuning for chest X-ray report generation. Furthermore, zero-shot MedFlamingo outperforms zero-shot Open-Flamingo on all metrics evaluated, showcasing the benefits of in-domain pre-training. Training from scratch outperforms the zero-shot performance of OpenFlamingo and MedFlamingo, which is remarkable since the whole representation alignment between the image and language modality is solely done with the MIMIC-CXR data. However, the fine-tuned models achieve the best scores across the board, presumably because they build upon their already aligned representations from their pre-training. This is also supported by pre-trained models reaching minimal validation loss in half the epochs compared to the model trained from scratch. Overall, MedFlamingo performs best across all evaluated metrics except for the BERTScore. This indicates that the in-domain pre-training facilitates the down-stream fine-tuning capabilities.

In conclusion, we want to highlight the benefits of using a diverse and already domain-specific pre-trained model like MedFlamingo as a basis for fine-tuning to a specialized clinical downstream task.

---

3. https://github.com/huggingface/evaluate

4. https://github.com/rajpurkarlab/CXR-Report-Metric & https://pypi.org/project/radgraph

## Acknowledgments

This work was supported by the Helmholtz Association's Initiative and Networking Fund on the HAICORE@FZJ partition.

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
