# OpenReview forum: "From Generalist to Specialist: Incorporating Domain-Knowledge into Flamingo for Chest X-Ray Report Generation"
_MIDL.io/2024/Short_Papers — MIDL 2024 Short Papers_

### Official Review · Reviewer_HDDf · 2024-04-25

**Confidence:** 4
**Final Rating:** 4

**Review:**

The paper "From Generalist to Specialist: Incorporating Domain-Knowledge into Flamingo for Chest X-Ray Report Generation" presents a thorough evaluation of the effectiveness of OpenFlamingo and MedFlamingo in generating radiological reports from chest X-ray images. The study focuses on the benefits of domain-specific fine-tuning in enhancing automated radiological report generation, emphasizing the importance of pre-training pathways for specialized medical tasks.

Merits:

Relevance: The paper addresses a significant challenge in medical image computing - automating the generation of accurate and reliable radiological reports from chest X-ray images.
Methodology: The study utilizes the MIMIC-CXR dataset and considers lexical and radiology-specific metrics for evaluation, providing a comprehensive analysis of the models' performance.
Results: The findings demonstrate the superiority of fine-tuned models over zero-shot models, highlighting the effectiveness of domain-specific pre-training in improving performance.
Contribution: The paper contributes valuable insights into the benefits of using domain-specific pre-trained models for fine-tuning in specialized clinical tasks, paving the way for more accurate diagnostic workflows.

Limitations:

Scope: The paper focuses specifically on chest X-ray report generation, limiting the generalizability of the findings to other medical imaging modalities.
Evaluation Metrics: While the study considers a range of evaluation metrics, the discussion could benefit from a deeper analysis of the implications of the results on clinical practice.
Comparison: A more detailed comparison with existing state-of-the-art methods in radiological report generation could provide additional context for the study's contributions.

---

### Decision · Program_Chairs · 2024-04-26

Accept